# ELISA- and Activity Assay-Based Quantification of BMP-2 Released *In Vitro* Can Be Biased by Solubility in “Physiological” Buffers and an Interfering Effect of Chitosan

**DOI:** 10.3390/pharmaceutics13040582

**Published:** 2021-04-19

**Authors:** Julius Sundermann, Steffen Sydow, Laura Burmeister, Andrea Hoffmann, Henning Menzel, Heike Bunjes

**Affiliations:** 1Technische Universität Braunschweig, Institut für Pharmazeutische Technologie und Biopharmazie, 38106 Braunschweig, Germany; j.sundermann@tu-braunschweig.de; 2Technische Universität Braunschweig, Institut für Technische Chemie, 38106 Braunschweig, Germany; s.sydow@tu-braunschweig.de (S.S.); h.menzel@tu-braunschweig.de (H.M.); 3Niedersächsisches Zentrum für Biomedizintechnik, Implantatforschung und Entwicklung (NIFE), 30625 Hannover, Germany; Laura.Burmeister@gmx.net (L.B.); Hoffmann.Andrea@mh-hannover.de (A.H.); 4Laboratory of Biomechanics and Biomaterials, Department of Orthopedic Surgery, Graded Implants and Regenerative Strategies, Hannover Medical School, 30625 Hannover, Germany; 5Technische Universität Braunschweig, Zentrum für Pharmaverfahrenstechnik–PVZ, 38106 Braunschweig, Germany

**Keywords:** bone morphogenetic protein 2, BMP-2, chitosan, release quantification, matrix effects

## Abstract

Chitosan nanogel-coated polycaprolactone (PCL) fiber mat-based implant prototypes with tailored release of bone morphogenic protein 2 (BMP-2) are a promising approach to achieve implant-mediated bone regeneration. In order to ensure reliable *in vitro* release results, the robustness of a commercially available ELISA for *E. coli*-derived BMP-2 and the parallel determination of BMP-2 recovery using a quantitative biological activity assay were investigated within a common release setup, with special reference to solubility and matrix effects. Without bovine serum albumin and Tween 20 as solubilizing additives to release media buffed at physiological pH, BMP-2 recoveries after release were notably reduced. In contrast, the addition of chitosan to release samples caused an excessive recovery. A possible explanation for these effects is the reversible aggregation tendency of BMP-2, which might be influenced by an interaction with chitosan. The interfering effects highlighted in this study are of great importance for bio-assay-based BMP-2 quantification, especially in the context of pharmaceutical release experiments.

## 1. Introduction

The bone-inducing growth factor bone morphogenetic protein 2 (BMP-2), which is primarily investigated for the repair of critical sized bone defects, has been the subject of many *in vivo* and *in vitro* studies [1]. In view of a lack of standardization of *in vitro* release conditions for implants and a frequently low *in vitro–in vivo* correlation (IVIVC) of release data, the question may arise as to the significance of *in vitro* release experiments for biomedical engineering. However, *in vitro* release studies are still of decisive importance both for the definition of critical material attributes during product development and for the subsequent quality control of drug-releasing implants [2]. Basic knowledge about the *in vitro* release is crucial for the further development of prototypes and a prerequisite for ethically justifiable *in vivo* experiments. 

When designing *in vitro* release setups it is important to adequately consider physiological parameters such as temperature, pH and osmolarity to ensure a certain degree of bio-relevance for these experiments [3]. Typically, release experiments during development of parenteral dosage forms are performed in phosphate buffered saline (PBS) [4]. Phosphate buffer is one of the main buffer components of blood and tissue fluid, but PBS has a completely different electrolyte composition and ionic strength than the interstitial fluid [5]. An alternative buffer that reflects the physiological electrolyte composition of the extracellular space is the so-called simulated body fluid (SBF) [6]. The high content of proteins and other macromolecules inside the extracellular space, which contributes to the formation of a so-called protein corona on surfaces [7], leads to macromolecular crowding [8] and has an influence on the surface desorption of proteins (Vroman effect) [9], is commonly not considered and mirrored during *in vitro* release testing.

Due to the high sensitivity requirements of *in vitro* release studies with BMP-2, which typically have to deal with BMP-2 amounts in the nanogram range, often only the enzyme-linked immunosorbent assay (ELISA) can be considered as quantification method. Besides the usually very low limit of quantification, ELISA offers the advantage of a very high specificity. ELISA allows distinctive quantification of structurally similar proteins such as BMP-2, vascular endothelial growth factor (VEGF) or insulin-like growth factor (IGF) within the same sample without mutual interference [10,11]. Certain biological activity tests such as the BRE-Luc assay provide a comparable specificity with an about 10 times lower sensitivity [12]. Superior to the ELISA, activity assays allow the determination of protein functionality.

For ELISA and other assays such as the BRE-Luc assay to provide reliable quantification results during *in vitro* release studies, certain conditions must be met in the sample buffer. Quantification by these assays is essentially based on a concentration-dependent protein–protein interaction between the dissolved protein and a specific antibody or BMP receptors, respectively. This implies, as we assume, that two critical conditions are met, which were both subject of this study: Firstly, the sample has to be free of substances that can potentially interfere with the highly specific protein–protein interactions under investigation. Secondly, for a valid quantification, the protein must be molecularly dispersed with the same aggregation tendency in both the sample and the standard [13]. In case the aggregation tendency differs between the sample and the standard, strongly increased and strongly decreased recoveries are both conceivable. It can be assumed that these aggregation-correlated recovery effects lead to a reduced recovery in case of an increased aggregation in the sample and to an increased recovery in the opposite case.

Especially in release experiments involving nanogram amounts of unlabeled BMP-2 and PBS as release medium, some inconsistencies like fluctuating BMP-2 concentrations [14] or a very low BMP-2 release in relation to the load quantity have been observed [15,16,17,18,19]. Some authors have reported different BMP-2 release rates in PBS as compared to cell culture medium or serum [11,20]. The overall aim of this study was to gain a better understanding of the interfering effects on ELISA- and activity assay-based quantification of BMP-2 observed during previous *in vitro* release studies and to carefully elaborate reaction procedures for reliable, reproducible release results.

BMP-2 has an isoelectric point (pI) of about 8.5 after chaotropic denaturation [21] and the tendency to form redissolvable aggregate particles of native protein with a pI of 7.7–8.3 at physiologic ionic strength and pH [22,23]. We were interested in the potential influence of poor BMP-2 solubility in physiological buffers like PBS [24] on the outcome of *in vitro* release tests. Furthermore, chitosan, which may be applied in controlled release systems for BMP-2 [25,26], was investigated as possible interfering substance. Although both, BMP-2 and chitosan, are positively charged at physiological pH, molecular dynamics simulations revealed an attractive interaction of these two substances [27]. Taking the two potential effects together, we wanted to explore whether the BMP-2 release as determined in *in vitro* studies was influenced by aggregation-correlated recovery effects in PBS-based buffers and SBF, especially in the presence of chitosan. 

## 2. Materials and Methods 

All chemicals were purchased from Sigma-Aldrich, Taufkirchen, Germany unless otherwise stated. 2-(*N*-morpholino) ethanesulfonic acid (MES; ≥ 99%) was purchased from Carl Roth, Karlsruhe, Germany. *E. coli* derived recombinant human bone morphogenetic protein 2 (rhBMP-2), simply referred to as BMP-2 in the following, was produced at the Institute for Technical Chemistry, Leibniz Universität Hannover, Hannover, Germany, as previously described [22].

### 2.1. Implant Prototypes

The implant prototypes are based on modified polycaprolactone (PCL) fiber mats which were described before [26]. The structure of the implant prototypes is shown schematically in Figure 1.

Briefly, the PCL fiber mats were electrospun from a PCL (M_n_ = 80,000 g mol^−1^) solution of 17 wt% in 2,2,2-trifluoroethanol at the Institute for Multiphase Processes at the Leibniz Universität Hannover, Germany, according to an optimized electrospinning method described before [29]. The electrospun fiber mats with fiber diameters of about 2.5 µm and a thickness of about 0.33 mm (microscopically determined as previously described [26]) were coated with a chitosan-graft-PCL (CS-g-PCL) copolymer to generate a positively charged interlayer as previously described by de Cassan et al. [29]. To obtain a negative surface charge, the CS-g-PCL coated fiber mats were additionally coated with an alginate layer (CS-g-PCL-Alg) by immersion in an aqueous solution of sodium alginate (5 mg mL^−1^) for 10 min. Subsequently, the fiber mats were rinsed several times with deionized water, vacuum-dried for 24 h at room temperature and cut into sections of 8 × 16 mm (final size of implant prototypes).

The fiber mat samples were coated with chitosan tripolyphosphate (CS-TPP) nanoparticles as previously described [26]. In order to obtain chitosan with the desired solubility properties, the degree of acetylation (DA) was adjusted. The general procedure of chitosan acetylation and nanoparticle preparation was published previously [30]. Briefly, purified CS with a DA of 17% was dissolved in 1% (*v/v*) acetic acid to get a concentration of 5 mg mL^−1^. After volumetric doubling with ethanol, acetic acid anhydride was added to the reaction mixture in an amount calculated according to Freier et al. [31]. The mixture was stirred for 18 h at room temperature. After subsequent dialysis against deionized water the acetylated CS was lyophilized. The DA was calculated from ^1^H-NMR spectra as 42%.

For particle preparation, the acetylated CS was dissolved in 0.1% (*v/v*) acetic acid at a concentration of 1 mg mL^−1^. Tripolyphosphate (TPP) was dissolved in deionized water at the same concentration. The TPP solution was rapidly mixed with the CS solution in a CS-TPP ratio of 3: 1, which in previous studies reproducibly resulted in the formation of nanoparticles with a z-average diameter in the range of 143 ± 2 nm and a zeta potential of about 27 ± 1 mV [28]. 

For BMP-2-containing nanoparticle suspensions, 100 µL of 0.132 mg mL^−1^ BMP-2 in 50 mM MES buffer (pH 5) were added to 900 µL CS solution before mixing with TPP to obtain 1 mL of nanoparticle suspension with a BMP-2 concentration of 13.2 µg mL^−1^ (Figure 2). Implant prototypes used for the release experiment with switch of medium type (cf. 2.2) were loaded using a BMP-2 concentration of 6.6 µg mL^−1^.

BMP-2-containing implant prototypes were produced by incubating stacks of four fiber mat samples in BMP-2-containing nanoparticle suspensions for 15 min, which in previous studies resulted in a BMP-2 loading of about 105 ± 16 ng per mg of fiber mat [26]. In order to produce implant prototypes without BMP-2, nanoparticle suspensions without BMP-2 were used accordingly.

### 2.2. In Vitro BMP-2 Release Experiments

Release experiments were performed in 1 mL of simulated body fluid (SBF) at pH 7.4 or in Dulbecco’s PBS supplemented with 0.05% Tween 20 and 0.1% BSA (PBS + Tween + BSA is simply referred to as PTB in the following, the PTB was part of the TMB ELISA Buffer Kit, Catalog Number 900-T00, Peprotech Inc., Rocky Hill NJ, USA). The pH of PTB was 7.2. The exact composition of SBF and PTB is given in Table 1.

Fiber mat samples were incubated with release media inside of 2 mL polypropylene Low-Binding Tubes (Sorenson BioScience, Salt Lake City, UT, USA, Cat# 12180) at 37 °C without agitation. 

In order to investigate the influence of the release media on the release kinetics in more detail, an additional release experiment with a change of the medium type after the initial burst release phase was performed. The implant prototypes were initially immersed in SBF and in the course of sampling after 24 h, the release medium was changed to PTB. The further progress of the release was compared with that of fiber mats that had been in PTB from the beginning.

All release experiments were carried out in triplicate. Sampling was performed at specified points in time over a period of one to four weeks. At each sampling time, fiber mat samples were removed from the tubes, blotted against clean paper towels to remove excess medium and transferred to fresh release medium. After removing the fiber mat samples, the tubes with the released BMP-2 were stored at 4 °C until quantification of released protein by ELISA or BRE-Luc assay.

In order to validate the BMP-2 release results, implant prototypes of identical composition that did not contain BMP-2 were extracted in PTB using exactly the same conditions as in the release experiments. 330 ng of BMP-2 was added to each of the extraction samples and the recovery of BMP-2 was determined by ELISA.

### 2.3. Quantification of BMP-2 Activity by BRE-Luc Assay

A mouse muscle satellite cell line C2C12 (Deutsche Sammlung für Mikroorganismen und Zellkulturen, Braunschweig, Germany, cat.-no. ACC 565) was stably transfected with the BMP Responsive Element (BRE) firefly luciferase reporter plasmid containing an inhibitor of the differentiation promoter luciferase construct from Korchynsky et al. [12]. The BRE Luciferase (BRE-Luc) assay was performed as previously described [32] using a 96 well-plate format as described in [22].

### 2.4. ELISA Quantification of BMP-2 

A commercial BMP-2 ELISA (BMP-2 TMB ELISA Development Kit, Peprotech Inc., Rocky Hill, NJ, USA) was used according to the manufacturer’s protocol with slight modifications. BMP-2 originating from the source described in the materials section was used as standard for the concentration calculation. Measurements were performed with an Infinite^®^ 200 pro plate reader (Tecan Ltd., Mänedorf, Zürich, Switzerland). Standard curves were fitted with OriginPro 2015 (Origin^®^ Labs, Northampton, MA, USA) using a 4PL-logistic curve fit.

## 3. Results

### Recovery of Released BMP-2

There was a high correlation between the release curves obtained by ELISA and BRE-Luc assay albeit with certain quantitative differences (Figure 3). In PTB, ELISA indicated sustained release of 158 ± 7 ng mg^−1^ over a period of 14 days with an initial burst effect. In contrast, a total release of only 1.3 ng mg^−1^ was detected by ELISA in SBF. By using the BRE-Luc assay no release at all could be measured in SBF. A similarly low release at the detection limit of the ELISA was also measured in PBS without addition of Tween and BSA (data not shown).

In order to distinguish whether BMP-2 release into the SBF medium was reduced or whether BMP-2 was released but could not be recovered from the release medium, an additional release test with a change from SBF to PTB after 24 h was performed (Figure 4).

As long as the release study was performed in SBF, no BMP-2 could be quantified by ELISA. However, after changing the medium to PTB, a BMP-2 release could be determined. The release rate of the samples during the first 7 h after medium switch from SBF to PTB was significantly higher (6.1 ± 1.5 ng mL^−1^ h^−1^) than that of the prototypes which had previously been released in PTB for the same time of 24 h (3.3 ± 0.1 ng mL^−1^ h^−1^). Compared to the burst release of 50.5 ± 8.7 ng mL^−1^ h^−1^ measured during the first 8 h of release in PTB, however, the release after changing the medium from SBF to PTB was about 8 times lower, indicating that high amounts of BMP-2, which could not be recovered by ELISA, had already been released into SBF.

To determine whether other components of the implant prototypes released into the release medium in addition to BMP-2 had an influence on ELISA recovery, these components were extracted from BMP-2-free implant prototypes using the same release setup. The recovery of 330 ng/mL BMP-2, which was spiked to the release medium samples containing the extracted implant components is shown in Figure 5.

The BMP-2 ELISA recovery was increased by a factor of about 3 in the first extraction sample and decreased continuously with the ongoing extraction until it reached approximately 100% after about 74 h. In order to determine which dissolving component of the implant prototypes caused the increased recovery, the possibly dissolving components of the fiber mats, chitosan (CS), tripolyphosphate (TPP), CS-TPP nanoparticles (CS-TPP-NP, mixture of CS and TPP with a CS-TPP ratio of 3: 1) and extracts from implant prototypes coated with alginate or with alginate + CS-TPP-NP were measured using ELISA. None of the components tested had a detectable effect on the blank value of the ELISA (Figure 6A).

In contrast, the recovery of 6.4 ng mL^−1^ BMP-2, which had been added to the component solutions, was clearly influenced by the different additives (Figure 6B). Apart from the extraction samples (last two columns), the added components were significantly more concentrated than in the release samples, which is reflected in the deviating pH value in these samples compared to PTB. The different pH value in these samples (CS-TPP-NP, CS and TPP) may have had an additional effect on the ELISA recovery.

Measurements in the extract from the alginate coated fiber mats (PCL + Alg (24 h)) showed a reduced BMP-2 recovery (57 ± 7%), while measurements in the extract from alginate + CS-TPP-NP coated fiber mats (PCL + Alg + NP (24 h)) indicated a strongly increased recovery (263 ± 19%). Accordingly, the increase in recovery was associated with CS or CS-TPP-NP. To investigate the influence of CS concentration on the recovery of BMP-2 in the ELISA, the recoveries of 4 and 8 ng/mL BMP-2 were determined in samples to which different amounts of chitosan (with or without TPP) had been added (Figure 7). Irrespective of whether TPP was additionally added, there was an increased recovery of approx. 272 ± 50% for 4 ng and of approx. 162 ± 30% for 8 ng BMP-2. No correlation between chitosan concentration or pH value and recovery was observed for the investigated chitosan concentrations between 21 and 333 ng mL^−1^.

To investigate the effect of chitosan on the ELISA more closely, BMP-2 concentration series diluted in PTB, which effectively contained 10 or 100 µg/mL CS-TPP-NP during the ELISA, were measured in comparison to the standard concentration series (Figure 8).

The extinction values obtained after performing the ELISA showed a different correlation to the BMP-2 concentration. The values obtained at zero or very low concentrations of BMP-2 were practically identical. However, the measured extinction increased more steeply with higher BMP-2 concentrations in samples that contained CS-TPP. The steeper correlation between BMP-2 concentration and measured extinction (increased sensitivity) was more pronounced in samples which contained more CS-TPP (B compared to C in Figure 8).

## 4. Discussion

The number of antigen-antibody reactions that occur between detection antibodies and the analyte is decisive for each ELISA result [33]. A prerequisite for 100% recovery is that this number of protein–protein interactions depends only on the analyte concentration. Altered ELISA recoveries in artificial sample matrices can be caused by so called matrix effects. Matrix effects can be explained by interactions between sample components, the analyte and/or the assay reagents. These interactions alter the response of the ELISA to the sample concentration compared to the standard series and therefore lead to false quantification results. Proteins (albumin, antibodies, etc.), lipids, carbohydrates, polyanions, small molecules and a different ionic strength are possible causes for matrix effects [34,35]. In the case of BMP-2, there seems to be a special matrix effect associated with the poor solubility and strong aggregation tendency of this protein. In PBS and other buffers with pH 7.4, BMP-2 forms insoluble aggregates which can be completely removed by centrifugation at 17,000× *g* [22]. The reduced recovery of BMP-2 after release into media buffered at physiological pH, such as SBF, therefore seems plausible. However, despite the poor solubility and the strong aggregation tendency of BMP-2, it is possible to determine the concentration of this protein using commercially available ELISA kits. The standard series of these ELISA kits are performed in PBS-based buffers at pH 7.2. It is therefore reasonable to assume that the release of BMP-2 in PBS or in the Tris- and phosphate buffer based SBF can also be tested and quantified by ELISA. However, as the comparison of the determined release curves in SBF and in PTB shows, this was not the case under the conditions employed in the current study. To obtain meaningful results it was crucial to add BSA and the surfactant Tween 20 to the release medium. Tween 20 and BSA apparently had a solubilizing effect on BMP-2 and prevented its precipitation at pH 7.2. Previous studies have shown that albumin influences the size of BMP-2 aggregates [23]. It was also found that BMP-2 still largely remained in the form of redissolvable aggregates despite the addition of BSA and Tween 20.

If the tendency for an analyte to aggregate in a sample is higher than in the reference, it can be assumed that assays based on protein–protein interaction, such as ELISA or cell culture assays, would generally provide reduced recoveries. On the other hand, in case aggregation takes place under the buffer conditions of the standard series, a reduction of the aggregation tendency in the sample may result in an increased recovery. The BMP-2 binding ability of sulfated polysaccharides such as heparin, heparan sulfate and dextran sulfate as well as of sulfated chitosan derivatives has been reported to potentiate BMP-2-induced osteogenic activity in cell culture-based assays [36,37,38]. A similar potentiation of ELISA response was observed upon addition of chitosan in the present study. A comparably high recovery associated with the addition of chitosan was also found by BRE-Luc assay [26]. The high correlation of the release curves obtained with the BRE-Luc assay and the ELISA shows that both assays were similarly biased by the effect of increased recovery (Figure 3).

Chitosan is known to bind BMP-2 although both molecules have an overall positive charge at neutral pH [27]. The chitosan binding may have a solubilizing effect on BMP-2 by preventing its ionic self-interaction (Figure 9).

Chitosan addition also increases the ionic strength of the solution, which further improves the solubility of BMP-2 [39].

Since BMP-2 forms reversible aggregates that grow in a concentration-dependent manner, solubilization of BMP-2 may have led to a reduction in the size of the aggregates and thus to an increase of the number of individual BMP-2 units. The increased number of BMP-2 units available for ELISA interaction might explain the increased sensitivity after addition of chitosan (Figure 7 and Figure 8). Quantification methods in which released BMP-2 can be detected irrespective of the state of aggregation, such as quantification via fluorescence- or radioactive labelling, or quantification methods in which the aggregates are completely degraded, such as selective reaction monitoring, should not be affected by this specific recovery issue.

## 5. Conclusions

Two lessons can be learned from this study. First, *in vitro* release studies with BMP-2 should not be performed at physiological pH without the presence of solubilizing additives because of the low solubility of this protein. For example, a mixture of 0.05% Tween 20 and 0.1% BSA in PBS, in the composition in which they are also used in commercially available ELISAs, appears to be suitable as release medium.

Second, the quantification of BMP-2 in media with solubilizing additives is still highly susceptible to aggregation-based matrix effects which, as shown for the polyelectrolyte chitosan, can lead to strongly increased recovery rates. When quantifying BMP-2 in the context of *in vitro* release studies with biodegradable implant prototypes, special attention should therefore be paid to the influence of the dissolving components of the implant prototypes on BMP-2 recovery in biological assays.

The interfering effects on BMP-2 quantification discussed in this study are considered to be relevant only for quantification methods based on protein–protein interactions.

## Figures and Tables

**Figure 1 pharmaceutics-13-00582-f001:**
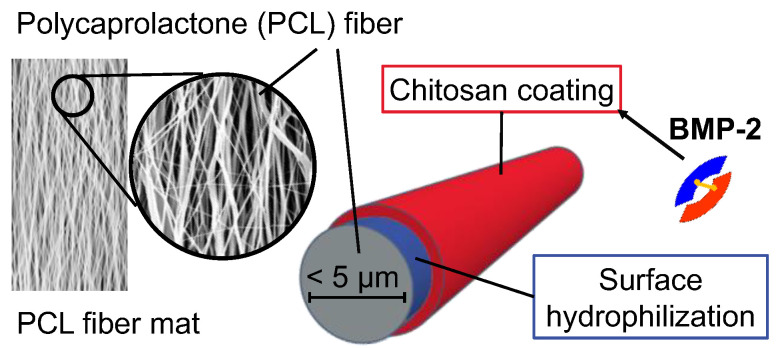
Schematic structure of the implant prototypes with a bone morphogenetic protein 2 (BMP-2)-containing chitosan coating. A detailed study on the coating mechanism and the resulting structure of the polyelectrolyte coating was performed by Sydow et al. [28].

**Figure 2 pharmaceutics-13-00582-f002:**
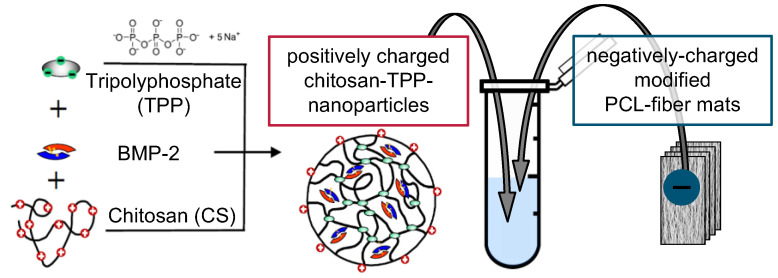
Chitosan coating/BMP-2 loading process of implant prototypes.

**Figure 3 pharmaceutics-13-00582-f003:**
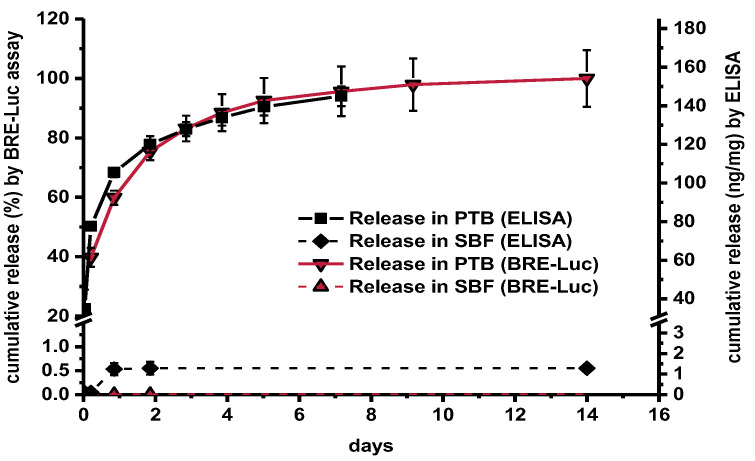
BMP-2 release from identical fiber mats in PTB and SBF (BRE-Luc assay, ELISA, *n* = 3, error bars indicate SD between individual release experiments).

**Figure 4 pharmaceutics-13-00582-f004:**
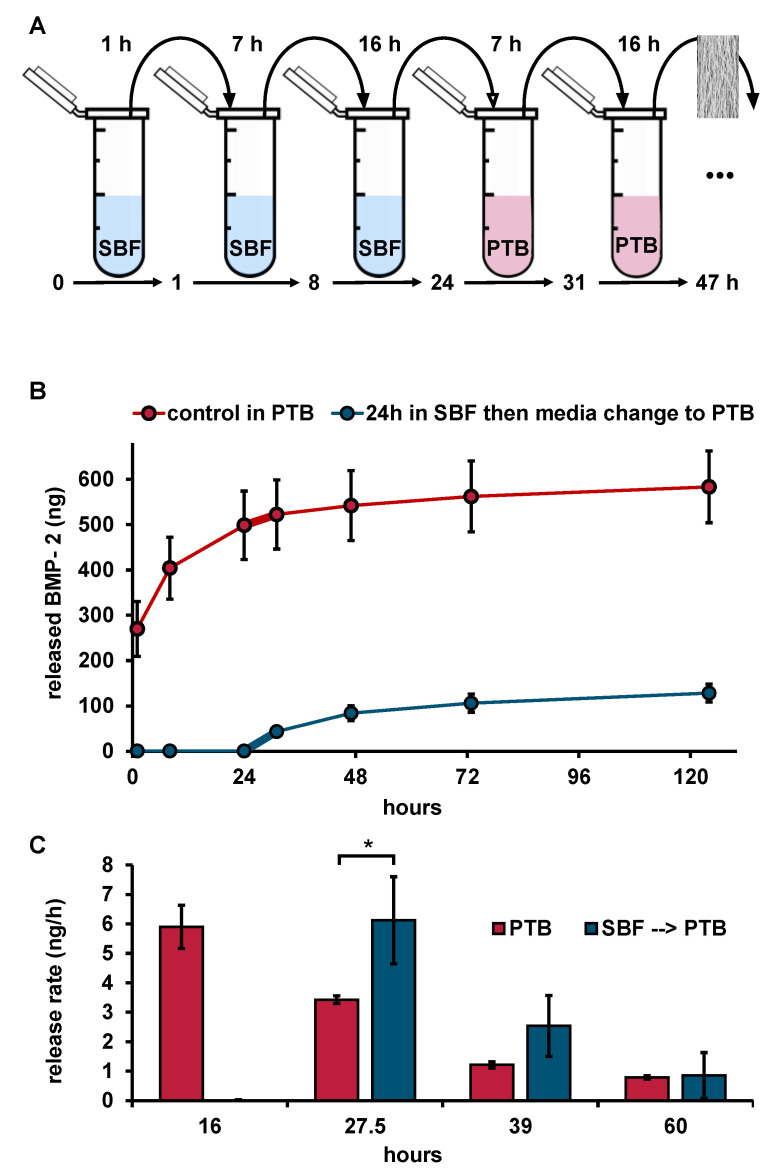
(**A**) Illustration of the release experiment with change of medium type. (**B**) Released BMP-2 (ELISA) from implant prototypes over time with and without medium switch (*n* = 3), thicker connection line highlights first release period after medium switch. (**C**) Release rates with and without medium switch (*n* = 3). Implant prototype mass: 10 mg each, error bars indicate SD between single release experiments. Statistics was performed using one sided t-test with the level of significance set at *p* ≤ 0.05 (* ≤ 5%).

**Figure 5 pharmaceutics-13-00582-f005:**
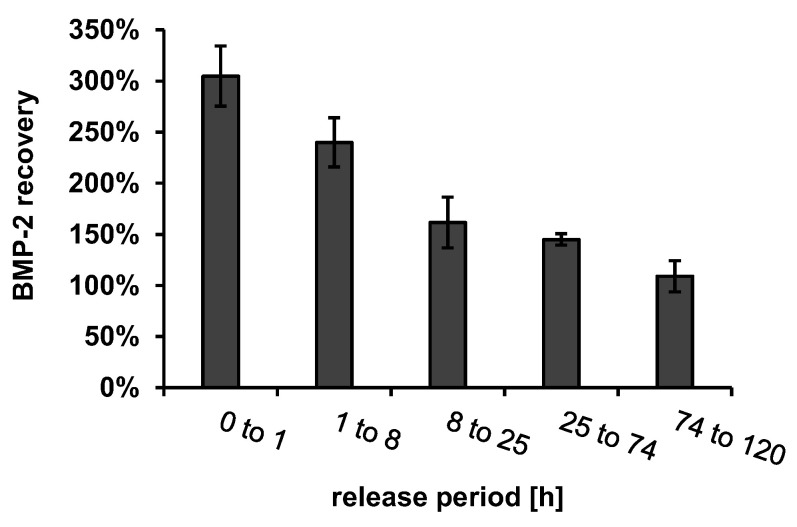
ELISA recovery of 330 ng mL^−1^ BMP-2 in release medium (PTB), containing dissolved components of the implant prototypes after extraction intervals, corresponding to the intervals of release experiments (*n* = 3, error bars indicate SD).

**Figure 6 pharmaceutics-13-00582-f006:**
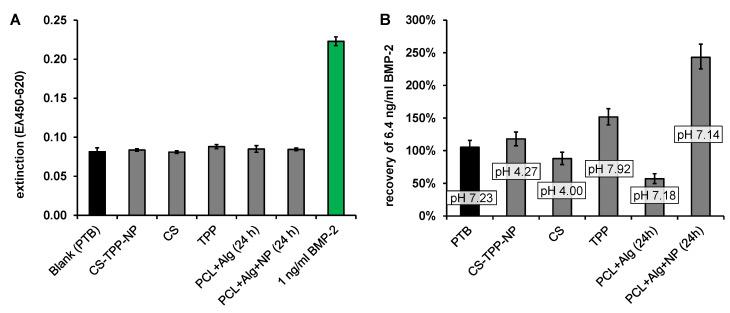
Effect of 0.3 mg mL-1 CS-TPP-NP/CS/TPP or an 24 h extract (release conditions) from an alginate coated (PCL-Alg) or an alginate and CS-TPP-NP coated (PCL + Alg + NP) implant prototype on: (**A**) the ELISA signal without BMP-2 (last column: signal of 1 ng/mL BMP-2 in PTB for comparison) and (**B**) the recovery of 6.4 ng/mL BMP-2 (*n* = 3, error bars indicate SD).

**Figure 7 pharmaceutics-13-00582-f007:**
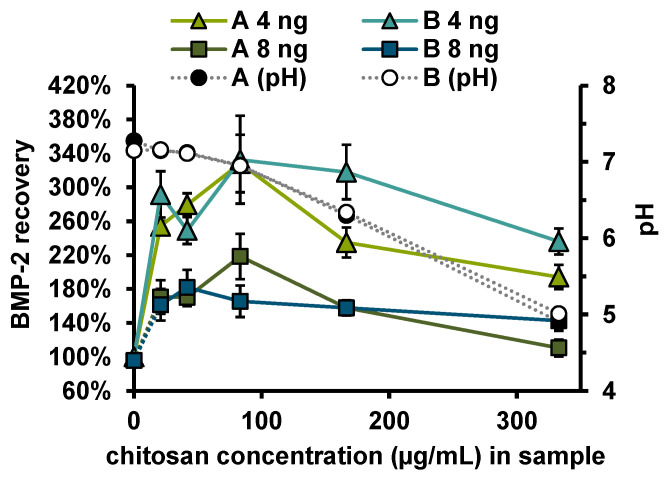
Recovery of 4 ng (*n* = 6) or 8 ng (*n* = 3) BMP-2 and pH in PTB after addition of different concentrations of (A) CS and (B) CS-TPP. Error bars indicate SD.

**Figure 8 pharmaceutics-13-00582-f008:**
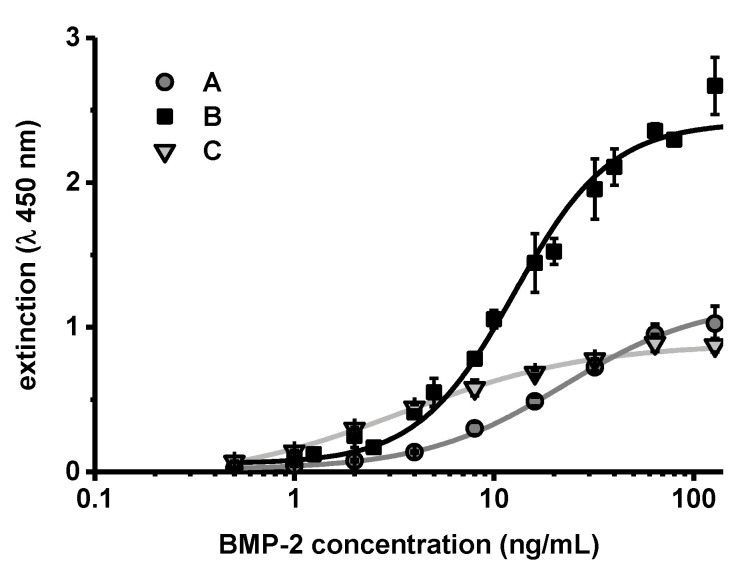
Extinction values obtained after BMP-2 ELISA of concentration series in (**A**) PTB, (**B**) PTB with 100 µg mL^−1^ chitosan (CS) or (C) PTB with 10 µg mL^−1^ CS (CS added as CS-TPP nanoparticles in (**B**,**C**)). The lines show corresponding four parameter logistic (4PL) regressions (*n* = 3, error bars indicate SD).

**Figure 9 pharmaceutics-13-00582-f009:**
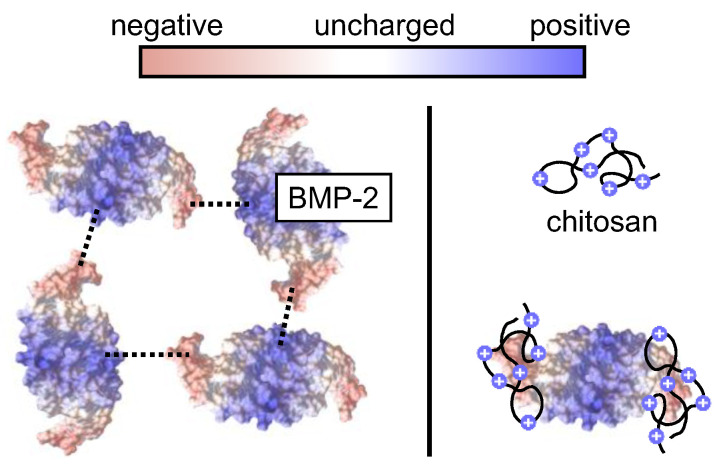
Schematic presentation of the proposed effect of chitosan on the aggregation tendency of BMP-2. Left (without chitosan): Ionic BMP-2 self-interaction, leading to BMP-2 aggregation [23]. Right: Binding of chitosan to the negatively charged tips of BMP-2 [27], potentially leading to a decreased aggregation tendency.

**Table 1 pharmaceutics-13-00582-t001:** Composition (mM) of release media. Simulated body fluid (SBF), Phosphate buffered saline supplemented with tween and bovine serum albumin (PTB)

Content	SBF	PTB
Na^+^	142.0	156.9
K^+^	5.0	4.4
Mg^2+^	1.5	-
Ca^2+^	2.5	-
Cl^−^	125.0	139.6
HCO^−^	27.0	-
HPO_4_^2−^	1.0	11.8
SO_4_^2−^	0.5	-
Tris	50.0	-
BSA	-	0.015
Tween 20	-	0.4

## Data Availability

All relevant data is contained in the article.

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
