# Peer review of "ELISA- and Activity Assay-Based Quantification of BMP-2 Released In Vitro Can Be Biased by Solubility in “Physiological” Buffers and an Interfering Effect of Chitosan"

_pharmaceutics, 2021, doi:10.3390/pharmaceutics13040582_

Round 1
Reviewer 1 Report
The authors report interesting sets of data on the recovery of BMP-2 in the different release media evaluated by ELISA. The authors might have already evaluated solubility and matrix effects in their in vitro release samples in different media. Therefore, in the authors’ previous paper (reference 26), the release experiments were performed with PTB. Then, in this paper the authors report its scientific backgrounds. For appropriately performing the release experiments, the experimental conditions should be carefully selected according to the objectives of the release study. However, with the effort alone made to design experimental conditions for in vitro release study, the paper is not considered as being suitable for publication because the final experimental conditions were already applied in the in vitro release of BMP-2 loaded in the same implant (i.e., reference 26). In additions, the authors are requested to address the following issues.
- The title of the paper is not suitable as it did not correctly reflect the study performed in the paper. The release data were biased but the authors revealed its reason and backgrounds. Therefore, more definitive keywords can be used for the title of the paper.
- There are numerous typographical errors. Please thoroughly check and correct them.
- Fig. 1 does not seem to demonstrate the exact coating structures mentioned in the method section of “Implant prototypes”.
- The dimensions of the final implant prototype need to be provided.
- Reference 26 is now published. Please add bibliographic information.
- Statistical analysis should be performed wherever possible.
- In Table 1, pH and ionic strength values of each release medium measured after preparation need to be added.
- Did author check the impact of experimental conditions on the biodegradation of PCL?
Reviewer 2 Report
- Why authors used tripolyphosphate (TPP) to modified chitosan?
- There are many “Error! Reference source not found.” in the manuscript.
- Why the release experiment need change medium type?
- Authors should explain what is “aggregation-correlated recovery effects” of BMP2 in Introduction section, how important is in the study? Why you want to study this effect? otherwise it is difficult to understand.
Reviewer 3 Report
The manuscript entitled „In-vitro release studies with BMP-2 can be biased by solubility and matrix effects” studies a chitosan nanogel-coated PCL fiber mat-based implant prototypes with tailored release of bone morphogenic protein 2 (BMP-2) which can be promising approach to achieve implant-mediated bone regeneration. The study supports in vitro release studies with BMP-2 should not be performed at physiological pH without the presence of solubilizing additives because of the low solubility of this protein, rather in a mixture of 0.05 % Tween 20 and 0.1 % BSA in PBS, in the composition in which they are also used in commercially available ELISAs, appears to be suitable as release medium. The manuscript is conscious, logically structured and well written. I only have few comments:
- Please check the reference in line 99, 132, 185, 218, 238, 250, 260, 267 and 313 because the reference source was not found.
- Line 106 describes the dimensions of electrospun PCL fiber mats with the diameters of about 2.5 μm and a thickness of about 0.33 mm. Does this filament size fit the particle size requirements of nano fibers or it is rather microparticle?
- Line 128 describes chitosan-TPP forms formed nanoparticles (nanogels) having an average hydrodynamic diameter of 143 ± 2 nm. In my opinion DLS is an appropriate method for investigation of nanoparticles, but in case of a hydrogel the result of DLS can be misleading, the hydrate shell of polymer depends on its swelling property. Therefore, I suggest to delete “nanogel” and use only simple “nanoparticle” in case of chitosan.
Round 2
Reviewer 1 Report
The authors convincingly answered the questions raised to them. Therefore, the revised manuscript is acceptable for publication without further modification.
Reviewer 2 Report
looks good